

# Microgeographic variation in body condition of three Mexican garter snakes in central Mexico

Erika Valencia-Flores[1], Crystian S. Venegas-Barrera[2], Victor Fajardo[3] and Javier Manjarrez[1]

[1] Facultad de Ciencias, Universidad Autónoma del Estado de México, Toluca, Estado de México, México
[2] Tecnológico Nacional de México/Instituto Tecnológico de Ciudad Victoria, Ciudad Victoria, Tamaulipas, México
[3] Facultad de Medicina Veterinaria y Zootecnia, Universidad Autónoma del Estado de México, Toluca, Estado de México, Mexico

## ABSTRACT

**Background**. Geographic variation in body size and condition can reveal differential local adaptation to resource availability or climatic factors. Body size and condition are related to fitness in garter snakes (*Thamnophis*), thus good body condition may increase survival, fecundity in females, and mating success in males. Phylogenetically related species in sympatry are predicted to exhibit similar body condition when they experience similar environmental conditions. We focused on interspecific and geographical variation in body size and condition in three sympatric Mexican garter snakes from the highlands of Central Mexico.

**Methods**. We assessed SVL, mass, and body condition (obtained from Major axis linear regression of ln-transformed body mass on ln-transformed SVL) in adults and juveniles of both sexes of *Thamnophis eques*, *T. melanogaster*, and *T. scalaris* sampled at different locations and ranges from 3–11 years over a 20-year period.

**Results**. We provide a heterogeneous pattern of sexual and ontogenic reproductive status variations of body size and condition among local populations. Each garter snake species shows locations with good and poor body condition; juvenile snakes show similar body condition between populations, adults show varying body condition between populations, and adults also show sex differences in body condition. We discuss variations in body condition as possibly related to the snakes' life cycle differences.

# INTRODUCTION

Organisms usually respond to differences in environmental conditions by exhibiting local adaptation in phenotypic traits. Geographic variation in phenotypic traits associated with body size and condition can reveal differential adaptation of local populations to local biotic and abiotic fluctuations as presence of related species, resource availability, or climatic factors (*Bronikowski & Arnold, 1999*; *Bronikowski, 2000*; *Miller et al., 2011*). Also, geographic variation in body size and body condition can reveal fundamental variation

Corresponding author
Javier Manjarrez, jsilva@uaemex.mx

in selective pressures, especially in reptiles such as snakes (*Bronikowski & Arnold, 1999*; *Miller et al., 2011*). Thus, analyses of geographic variation in body size and condition are important to explain locally variable adaptations that produce morphological diversity in snake species. Geographic variation in body condition comes from many causes, including phenotypic plasticity (*Krause, Burghardt & Gillingham, 2003*) or microevolutionary change among natural populations (*Bronikowski, 2000*). These population differences may arise from geographic variation in food resources (*Bronikowski & Arnold, 1999*), climate (*Ashton, 2001*), or intra-inter species interactions (e.g., *Kurzava & Morin, 1994*).

Body condition is an expression of weight and length (size-adjusted body mass), and it is correlated with body reserves (*Hayes & Shonkwiler, 2001*), especially with energy stores in the liver, muscle, and fat of snakes (*Bonnet et al., 1998*; *Falk, Snow & Reed, 2017*). During periods of low resource availability, starvation and low body reserves are a good predictor of mortality (*Shine et al., 2001*; *Kissner & Weatherhead, 2005*), decreased reproductive status (*Naulleau & Bonnet, 1996*; *Lind & Beaupre, 2015*; *Catherine, LeMaster & Lutterschmidt, 2018*), and low growth rates in snakes (*Bronikowski, 2000*).

Thus, there is a relation between body size and condition with fitness, but in different ways for the two sexes, especially with reproductive status of snakes. For example, a good body condition may be associated with enhanced survival of both sexes of garter snakes, greater fecundity in female garter snakes, and increased mating success for males (*Naulleau & Bonnet, 1996*); thereby, a reduction in body condition may reduce reproductive capacity (*Lind & Beaupre, 2015*). Conversely, adult female snakes in poor condition that are carrying eggs experience greater mortality (*Madsen & Shine, 1993*; *Brown & Weatherhead, 1997*; *Shine et al., 2001*).

Additionally, phylogenetically related species in sympatry are predicted to exhibit similar body condition when they have similar ecology, because they share similar evolutive history, interspecific interactions and selective pressures (i.e., *Yom-Tov & Geffen, 2006*; *Koyama et al., 2015*; *Sivan et al., 2015*). For example, closely related species of garter snakes with highly overlapping ranges in Mexico, *Thamnophis melanogaster* and *T. eques*, show similar patterns of neonate body condition as a function of date of birth (*Manjarrez & San-Roman-Apolonio, 2015*).

To understand the complex evolution of body condition, we studied interspecific and geographical variation in traits known to be associated with body condition in three sympatric Mexican garter snakes (*Thamnophis* sp.) inhabiting five sites from the highlands of Central Mexico. Given that the geographic distribution of these three garter snakes comprises a range of different environmental conditions, we hypothesized that traits associated with body condition of snakes would potentially reveal a pattern of geographical variation among local populations that could be influenced by dietary differences, ontogenic reproductive status (juvenile, adult), and sex of snakes. We predicted that the geographic variation in body condition in garter snakes is influenced by diet differences among populations, such that body condition would vary among populations. We discuss possible body condition differences as they are related to life cycle differences.

In this study we assessed snout-vent length (SVL), mass, and body condition in adults and juveniles of both sexes from three sympatric garter snakes in the Central Mexican

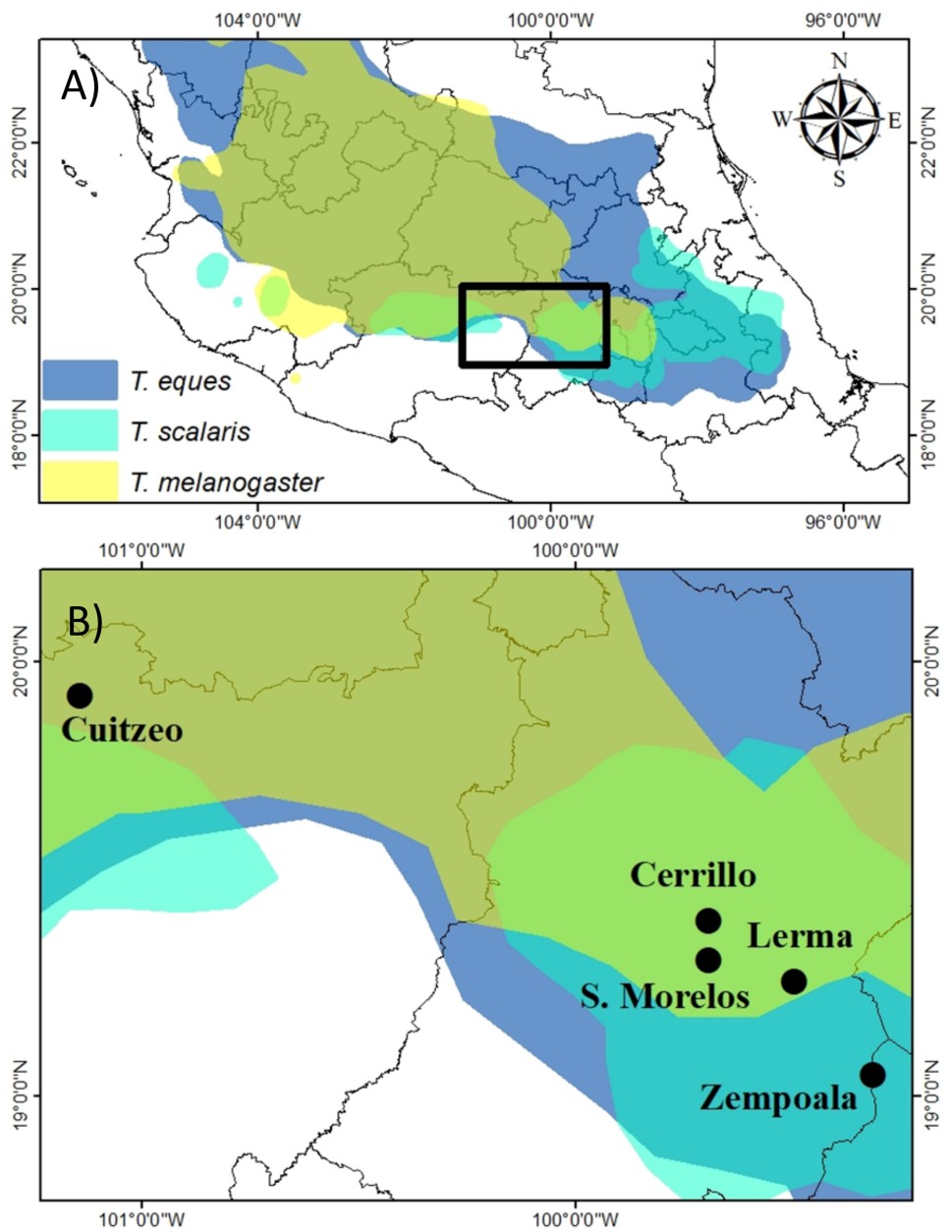

**Figure 1** (A) Geographic distribution of *T. eques, T. melanogaster* and *T. scalaris* in Central Mexico (digitalized from *Rossman, Ford & Seigel, 1996*) and (B) study locations in the Rio Lerma drainage.

Highlands (Fig. 1); Mexican garter snake (*Thamnophis eques*), Mexican Black-bellied garter snake (*T. melanogaster*), and Longtail Alpine garter snake (*T. scalaris*). They are grouped within the well-supported clade of garter snakes composed of species found mostly in Mexico (*Queiroz, Lawson & Lemos-Espinal, 2002*; *Guo et al., 2012*; *McVay & Carstens, 2013*). Garter snakes are the most abundant snake genus in Central Mexico (*Flores-Villela, Canseco-Márquez & Ochoa-Ochoa, 2010*). Garter snakes exhibit substantial variation within

and among species in certain aspects of morphology, behavior and physiology, a pattern specially demonstrated in two North American species recognized as examples of evolution (*Burghardt & Schwartz, 1999*). There is considerable intra-interspecific variation in color patterns, body size, diet, habitat, resistance to toxic prey, reproductive characterstcs and behavior that the patterns of the constraints may vary among poplations (*Burghardt & Schwartz, 1999*; *Rossman, Ford & Seigel, 1996*). In general, garter snakes are sexually dimorphic in body size (*Shine, 1993*) with females regularly larger than males (*Shine, 1994*). Almost all studies comparing the body condition of garter snake species were conducted separately for each sex and rarely have been combined in a single study; therefore, there is scarce information of possible sex differences in garter snake body condition, but see (*Rollings et al., 2017*).

We chose the species *T. eques*, *T. melanogaster*, and *T. scalaris* that occur in Central Mexico because there are no studies that describe the body condition or its possible interspecific or spatial variations under natural conditions for these three species. Only one study of *T. melanogaster* and *T. eques* detected body condition patterns in offspring born from females caught in the wild (*Manjarrez & San-Roman-Apolonio, 2015*). For both species, body condition of neonates differed by being lower in the early season and higher in the late season. Snout-vent length of neonates and mean mass of neonates per litter did not change throughout the birth season (*Manjarrez & San-Roman-Apolonio, 2015*).

*Thamnophis eques* is widely distributed from Central Mexico to southern New Mexico and Arizona in the United States (*Rossman, Ford & Seigel, 1996*). It is a generalist snake because it preys on both terrestrial and aquatic prey such as frogs, fish and tadpoles, and occasionally, mice and lizards (*Drummond & Macías García, 1989*; *Manjarrez, 1998*; *Manjarrez, Pacheco-Tinoco & Venegas-Barrera, 2017*). *Thamnophis melanogaster* is endemic to the Central Mexican Plateau. It is a semiaquatic snake present at the edge of water bodies and preys mostly on tadpoles, fish, and leeches (*Rossman, Ford & Seigel, 1996*; *Manjarrez, Macías García & Drummond, 2013*). *Thamnophis scalaris* is also endemic to Central Mexico (*Rossman, Ford & Seigel, 1996*). It inhabits forests and grasslands, where it specializes on earthworms, although it can eat vertebrates such as lizards and mice (*Manjarrez, Venegas-Barrera & García-Guadarrama, 2007*).

## MATERIALS & METHODS

In Central Mexico, we irregularly sampled garter snakes at eight different locations in the Rio Lerma drainage (Fig. 1A) over a period of 20 years, however, we selected only those five populations (Fig. 1B) with more than 24 records of snakes, which allowed us to make spatial and sex comparisons. We selected the records of snakes collected over three different years for *T. scalaris* (2003, 2005, and 2010) at three locations; seven years for *T. melanogaster* (2005–2011), at two locations, and eleven years for *T. eques* (2000–2003, 2005–2011) at three locations (Table 1). Locations are separated by 92.6 Km of mean distance (SD = 80 Km, range 9.5—215 km). Among the five sites, mean annual temperature ranged from 13.7°–18.1 °C and mean annual precipitation ranged from 116 mm–755.8 mm (Table 1). *Thamnophis eques* were captured between March and November, *T. melanogaster* between January and December, and *T. scalaris* between June and November.

**Table 1 Capture locations of *T. eques, T. melanogaster* and *T. scalaris* in Central Mexico.**

| Locality | Garter snake present | Coordinates N, W (Datum WGS84) | Elevation (m) | Mean annual temperature (°C) | Mean annual precipitation (mm) |
|---|---|---|---|---|---|
| Lerma, Estado de México | *T. eques, T. melanogaster, T. scalaris* | 19°14′28.73″, 99°29′41.14″ | 2,573 | 15.8 | 158.7 |
| Cerrillo, Estado de México | *T. eques* | 19°24′20.86″, 99°41′41.05″ | 2,550 | 13.7 | 116 |
| S. Morelos, Estado de México | *T. scalaris* | 19°18′49.58″, 99°41′29.07″ | 2,750 | 13.8 | 746.9 |
| Cuitzeo, Michoacan | *T. eques, T. melanogaster,* | 19°55′32.83″, 101°08′26.78″ | 1,837 | 18.1 | 755.8 |
| Zempoala, Morelos | *T. scalaris* | 19°02′53.40″, 99°18′44.54″ | 2,800 | 14.2 | 514 |

We found snakes by searching under rocks and tree trunks, and some were found simply basking on the ground. All snakes were captured by hand. Adult females were carefully examined for the presence of embryos, and those identified as gravid were excluded from analysis. Measurements of captured snakes included sex (visual inspection of tail-base breadth or by everting the male hemipenes in small snakes), snout-vent length (SVL), and mass (measured on an electronic scale ($\pm 0.1$ g)). Dietary differences among the localities were examined by analysis of stomach contents from *T. eques* and *T. melanogaster*. We obtained stomach contents by making the snakes regurgitate by abdominal palpation (*Fitch, 1987*). For *T. scalaris*, no stomach contents were recorded. Immediately after processing, snakes were released where they had been captured.

## Analysis

Individual body condition was calculated using residuals from the Major axis (MA) linear regression of ln-transformed body mass on ln-transformed SVL. This MA residual index is considered an excellent estimator of true snake body condition because it shows a strong association with body fat mass but not SVL (*Falk, Snow & Reed, 2017*). The condition that the MA linear regression is unbiased in with respect to size is considerable for hypothesis testing, because an absence of correlation with size permit to compare MA residual index across individuals of different size ranges. Particularly, only 2% of the variation in the MA residual index is associated with SVL (*Falk, Snow & Reed, 2017*). This regression was significant for all species (*T. eques*, $r = 0.90$, $P < 0.0001$; *T. melanogaster*, $r = 0.93$, $P < 0.0001$; *T. scalaris*, $r = 0.95$, $P < 0.0001$). Residuals were used to categorize body condition, with positive residuals corresponding to individuals with good body condition and negative residuals corresponding to individuals with poor body condition (*Weatherhead & Brown, 1996*; *Falk, Snow & Reed, 2017*). In this way, the average condition by location is interpreted as good or bad condition by location for each species.

Because the optimal body condition should approximate the true body condition of the snakes and should be unbiased with respect to body size, we evaluated this relationship with Kendall rank correlation coefficient to test for a correlation between body condition and ln-transformed SVL as a measure of size and estimated the percent variation in body

**Table 2** ANOVA of ln-SVL and ln-mass as dependent variables among locations, years and sex for each garter snake species.

|  | Location | Year | Sex |
|---|---|---|---|
| *T. melanogaster* | | | |
| body condition | 1.25 | 3.76** | 1.54 |
| SVL | 21.58*** | 6.56*** | 0.18 |
| mass | 29.21*** | 5.50*** | 0.00 |
| *T. eques* | | | |
| body condition | 5.59* | 2.56* | 0.21 |
| SVL | 12.08*** | 22.75*** | 0.99 |
| mass | 7.47** | 20.82*** | 1.66 |
| *T. scalaris* | | | |
| body condition | 14.06*** | 14.32*** | 0.23 |
| SVL | 7.12** | 0.34 | 0.67 |
| mass | 2.42 | 3.73* | 1.10 |

Notes.
*$P < 0.05$.
**$P < 0.001$.
***$P < 0.0001$.

**Table 3** Sex ratio (male:female) of *T. eques*, *T. melanogaster*, and *T. scalaris* for each population collected from Central Mexican Highlands ($df = 1$ for all tests).

|  | *T. eques* | | *T. melanogaster* | | *T. scalaris* | |
|---|---|---|---|---|---|---|
|  | Sex ratio | $\chi^2$ test ($P$) | Sex ratio | $\chi^2$ test ($P$) | Sex ratio | $\chi^2$ test ($P$) |
| Lerma | 1:1 | 0.45 (0.49) | 1:1 | 2.0 (0.15) | 1:1.5 | 12.46 (0.0004) |
| Cuitzeo | 1:2 | 4.33 (0.03) | 1:1.4 | 14.9 (0.0001) | | |
| Cerrillo | 1:3 | 6.76 (0.009) | | | | |
| S. Morelos | | | | | 1:1.7 | 9.94 (0.001) |
| Zempoala | | | | | 0:23 | 23.0 (<0.0001) |

condition. Also, with the coefficient of determination ($R^2$), we estimated the percent variation in body condition and mass that can be explained by SVL.

## Geographic comparison

The SVL and mass of snakes were transformed with natural logarithms prior to analyses. We utilized one-way analyses of variance (ANOVA) to compare body condition, SVL, and mass as dependent variables among populations of each species. In these analyses, we pooled male and female snakes because a three-way ANOVA (locality, year, and sex) indicated that body condition, SVL and mass within each species did not differ between sexes, but did differ among locality and between years (Table 2). We used a Chi-square goodness-of-fit test to determine if sex ratio among species was different than 1:1 (Table 3). Statistical significance was assessed at $\alpha = 0.05$. All data are reported as means $\pm 1$ *SD*.

## Sexual and size status comparison

Each snake was assigned an ontogenic reproductive status (juvenile, adult) according to size at capture (adult snakes >39.0, 33.0, and 34 cm SVL for *T. eques*, *T. melanogaster*

and *T. scalaris*, respectively; (*Manjarrez, 1998*; *Manjarrez, Venegas-Barrera & García-Guadarrama, 2007*). We performed a discriminant function analysis (DFA) for testing intraspecific differences (between location, sex, and size category) according to the mean of the exploratory variables (SVL, mass, and body condition) and for generating linear combinations that classify snakes as a function of their morphological traits associated with snake body condition. The grouping variables were location, sex (male, female) and ontogenic reproductive status. DFA is an inferential, descriptive multivariate procedure for testing differences between groups according to the mean of all variables and for generating linear combinations that classify objects as a function of their characteristics (Statistica, ver. 12; StatSoft, Inc., Tulsa, OK, USA).

The objective of DFA was to test differences between groups and identify which variables discriminate between two or more groups. Comparisons between groups were performed under the null hypothesis that morphological traits between categories of grouping variables were similar, and the estimated value was contrasted with the theoretical value of the F-distribution. We employed a probability of 0.05 to test the hypothesis, where P values lower than 0.05 were associated with groups of snakes showing different morphological traits, whereas values greater than or equal to 0.05 were associated with groups with similar morphological traits. The canonical average of the observations from each category (centroid) for the significant roots (canonical scores) was plotted, which reflects morphological variations between categories of grouping variables. The position of the centroids was interpreted using the variables that contributed most to discriminating between groups.

We chose those variables that exhibited a coefficient of the factor structure higher than 0.5 or lower than -0.5. The coefficients represent the correlation between the original variables and the roots. We applied one-way ANOVAs or Student-t with Statistica software (ver. 8.0; StatSoft, Tulsa, OK, USA) when only one morphological variable exhibited a coefficient of the factor structure higher than 0.5 or lower than –0.5.

This study received the approval of field permit (Secretaria del Medio Ambiente y Recursos Naturales #07164) and the ethics committee of the Universidad Autónoma del Estado de México (Number 4047/2016SF). All subjects were treated humanely on the basis of guidelines outlined by the American Society of Ichthyologists and Herpetologists (ASIH, 2004).

## RESULTS

The biggest species of garter snake was *T. eques* with a mean body size of SVL 43.43 ± 17.57 cm (range 12.51–81.30), mass of 55.62 ± 60.56 g (range 1.40–335.86, $n = 253$). *Thamnophis melanogaster* was slightly larger than *T. scalaris* (*T. melanogaster*: SVL 29.17 ± 41 cm [range 14.40–66.0], mass 19.10 ± 23.3 g [range 1.62–196.0], $n = 686$; *T. scalaris*: SVL 28.70 ± 9.21 cm [range 12.10–53.0], mass 16.44 ± 12.59 g [range 1.30–60.70], $n = 80$).

The number of males and females collected was independent of locations sampled for *T. melanogaster* ($\chi^2 = 0.001$, $df = 1$, $P = 0.97$), and *T. scalaris* ($\chi^2 = 3.69$, $df = 2$, $P = 0.15$), but dependent on location for *T. eques* ($\chi^2 = 10.4$, $df = 2$, $P = 0.006$). Considering all individuals collected, the sex ratio was biased toward females. For *T. eques* and *T. scalaris*,

**Table 4  Kendall rank correlation and $R^2$ coefficients of ln-mass and body condition on ln-SVL of *T. eques*, *T. melanogaster* and *T. scalaris*.**

| | *T. eques* $n = 253$ | $R^2$ | *T. melanogaster* $n = 686$ | $R^2$ | *T. scalaris* $n = 80$ | $R^2$ |
|---|---|---|---|---|---|---|
| Ln-mass | 0.77[*] | 0.84 | 0.81[*] | 0.88 | 0.80[*] | 0.92 |
| Body condition | −0.25[*] | 0.12 | −0.19[*] | 0.08 | −0.21[*] | 0.05 |

**Notes.**
[*]$P < .0001$.

the sex ratio was skewed toward females in two or three locations analyzed (Table 3), whereas the sex ratio for *T. melanogaster* was biased toward females in Cuitzeo but not in Lerma (Table 3). For *T. scalaris* the female bias was very distinct, especially Zempoala where no males were found (Table 3).

Both body condition (residuals from MA linear regression of ln-transformed body mass on ln-transformed SVL) and body mass were related to ln-SVL in each garter snake (Table 4). The $R^2$ values suggest that more than 80% of the variation in body mass is explained by SVL, and less than 12% of the variation in body condition is explained by SVL (Table 4).

## Geographic comparison
### *Thamnophis eques*
For the three locations that we analyzed for *T. eque* s (Lerma, Cerrillo and Cuitzeo), we observed a difference in mean body condition. *Thamnophis eques* from Lerma showed a mean poor body condition that was the lowest of the three populations ($F_{2,250} = 10.7$, $P < 0.0001$; Fig. 2), although snakes in this location were significantly larger than in the other two (ln-SVL $F_{2,250} = 6.7$, $P = 0.001$). Conversely, *T. eques* from Cuitzeo showed the best body condition, but the shortest length (Fig. 2). Mean body mass was not different between locations of *T. eques* (ln-mass $F_{2,250} = 2.2$, $P = 0.11$).

### *Thamnophis melanogaster*
For *T. melanogaster*, the statistical test did not detect a significant difference in mean body condition between the two locations, Lerma and Cuitzeo ($F_{1,684} = 3.1$, $P = 0.07$). However, the Lerma snakes were significantly larger (ln-SVL $F_{1,684} = 42.3$, $P < 0.0001$), and heavier than those collected in Cuitzeo (ln-mass $F_{1,684} = 56.4$, $P < 0.0001$; Fig. 2).

### *Thamnophis scalaris*
In this species the mean SVL and mass showed no differences among the three locations analyzed (Lerma, S. Morelos and Zempoala, ln-SVL $F_{2,77} = 1.55$, $P = 0.21$; ln-mass $F_{2,77} = 0.58$, $P = 0.56$), however, mean body condition was good in the individuals from S. Morelos and poor for those from Zempoala ($F_{2,77} = 20.9$, $P < 0.0001$; Fig. 2).

## Sexual and size status comparison
The results of DFA showed that each garter snake had a unique pattern of intraspecific differences. The body condition of juveniles *T. eques* was better than adults ($t_{227} = 43.3$, $P < 0.0001$). For *T. melanogaster* and *T. scalaris* the body condition of juveniles and adults was similar ($t_{684} = 0.01$, $P < 0.99$; $t_{78} = 1.5$, $P = 0.13$, respectively).

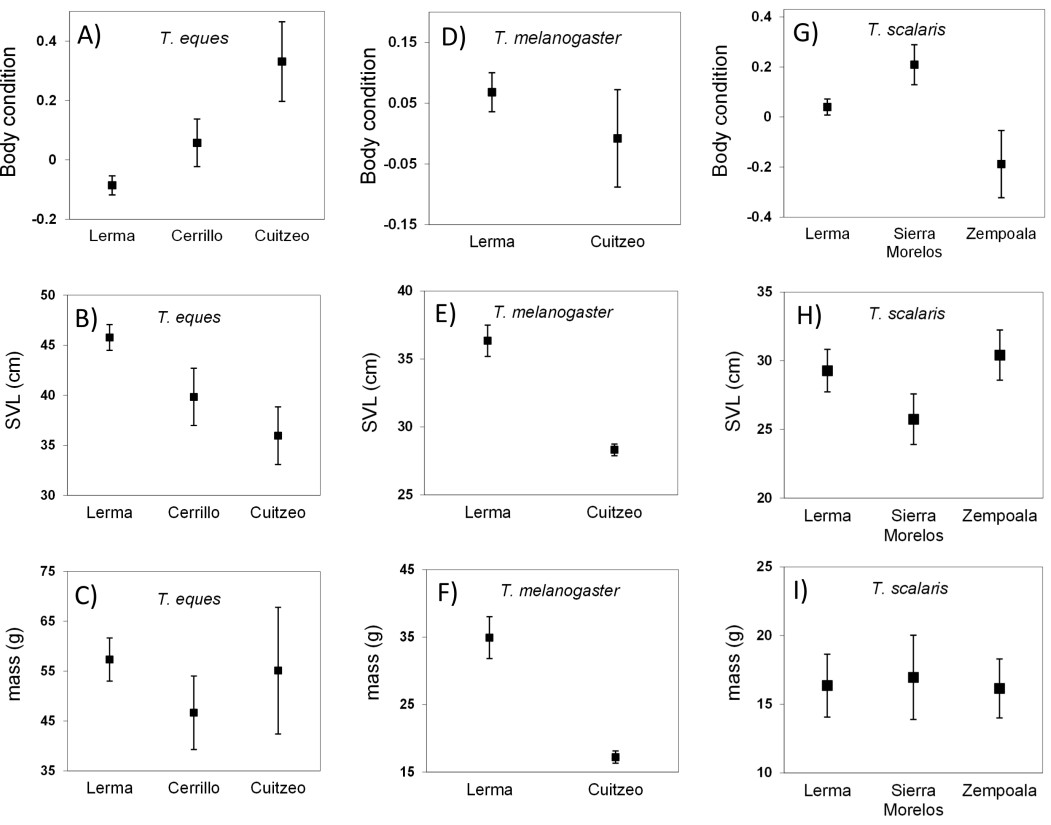

**Figure 2** **Body condition, SVL and mass (mean ± 1SE) of wild-caught snakes *T. eques* (A–C), *T. melanogaster* (D–F), and *T. scalaris* (G–I).** Snakes collected from locations in the Central Mexican Highlands over a period of 20 years. Body condition obtained of residuals from MA linear regression of ln-transformed body mass on ln-transformed SVL.

### Thamnophis eques

Juvenile females of Cuitzeo had a better body condition than juvenile females of Lerma ($t_{36} = 2.17$, $P = 0.03$), but body size (SVL and mass) were similar between Juvenile females of both locations ($F_{2,35} = 2.9$, $P = 0.06$). Juvenile males *T. eques* have similar body size and body condition between Lerma and Cuitzeo.

Adult males *T. eques* of Cuitzeo had a higher mass ($140 \pm 130.1$ g) than adult males of Lerma ($57.2 \pm 32.7$ g) and Cerrillo ($49.7 \pm 18.7$ g, ANOVA $F_{2,37} = 8.2$, $P < 0.0001$). Adult female *T. eques* of Lerma presented greater body size (SVL $59.0 \pm 9.3$ cm; mass $104.0 \pm 68.1$ g) than adult females of Cerrillo (SVL $49.7 \pm 8.5$ cm; mass $71.0 \pm 41.3$ g; DFA $F_{4,158} = 3.51$, $P = 0.008$, Fig. 3).

### Thamnophis melanogaster

Juvenile male *T. melanogaster* showed that body size traits and the body condition were similar between Lerma and Cuitzeo ($F_{1,165} = 1.3$, $P = 0.25$). In the case of juvenile female *T. melanogaster*, SVL was greater in Lerma ($26.2 \pm 4.8$ cm) than Cuitzeo ($23.3 \pm 4.7$ cm), and body condition was similar between both locations ($F_{1,260} = 5.06$, $P = 0.02$).

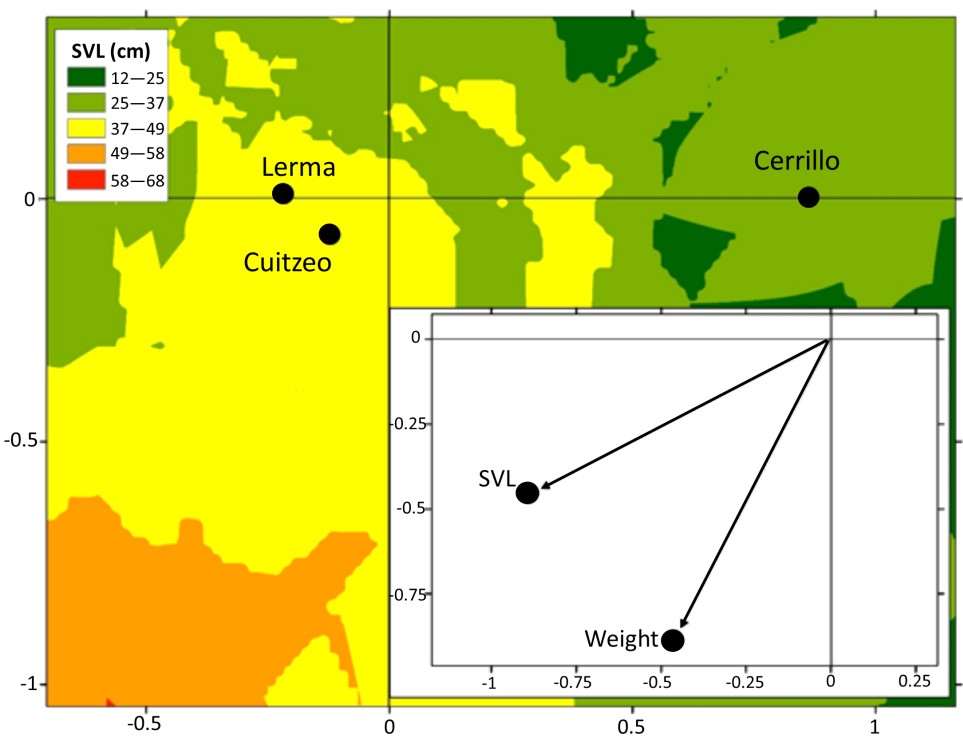

**Figure 3** Average canonical position (centroid) for Lerma, Cerrillo and Cuitzeo, obtained from a discriminant function analysis of body condition traits in adult female *T. eques* and factor structure. Isoclines represent variation on SVL of snakes in Lerma, Cerrillo and Cuitzeo.

Adult *T. melanogaster* of both sexes presented a similar pattern. A better body condition in Lerma than Cuitzeo (males: $0.09 \pm 0.29$ vs. $-0.04 \pm 0.24$; females $0.18 \pm 0.34$ vs. $-0.07 \pm 0.33$), and similar body size (SVL and mass) between Lerma and Cuitzeo (males: $F_{1,86} = 4.9$, $P = 0.02$; females: $F_{2,98} = 8.07$, $P = 0.0006$).

### Thamnophis scalaris

Only females *T. scalaris* (juvenile and adult) were enough to make comparisons between locations. Juvenile female *T. scalaris* of Zempoala were significantly longer, lighter, and had poor body condition than other locations. Lerma snakes showed lower SVL, mass, and average body condition, while snakes from Cerrillo and S. Morelos presented a better body condition, average SVL, and higher mass ($F_{1,165} = 1.3$, $P = 0.25$, Fig. 4A).

Adult female *T. scalaris* of Lerma and Zempoala had poorer body condition than those of Cerrillo and S. Morelos ($F_{6,62} = 8.4$, $P < 0.0001$, Fig. 4C).

### Stomach contents

Of the total snakes collected by species in this study, 17.4% (44 *T. eques*) and 13.3% (91 *T. melanogaster*) had some prey in the stomach. The diet of *T. eques* in Lerma and Cerrillo included aquatic prey (leeches, fish and tadpoles) and amphibious prey (frogs). The terrestrial prey (earthworms and mice) were only ingested by *T. eques* at Cerrillo (Table 5). At Cuitzeo, *T. eques* consumed mainly fish and only some leeches (Table 5).

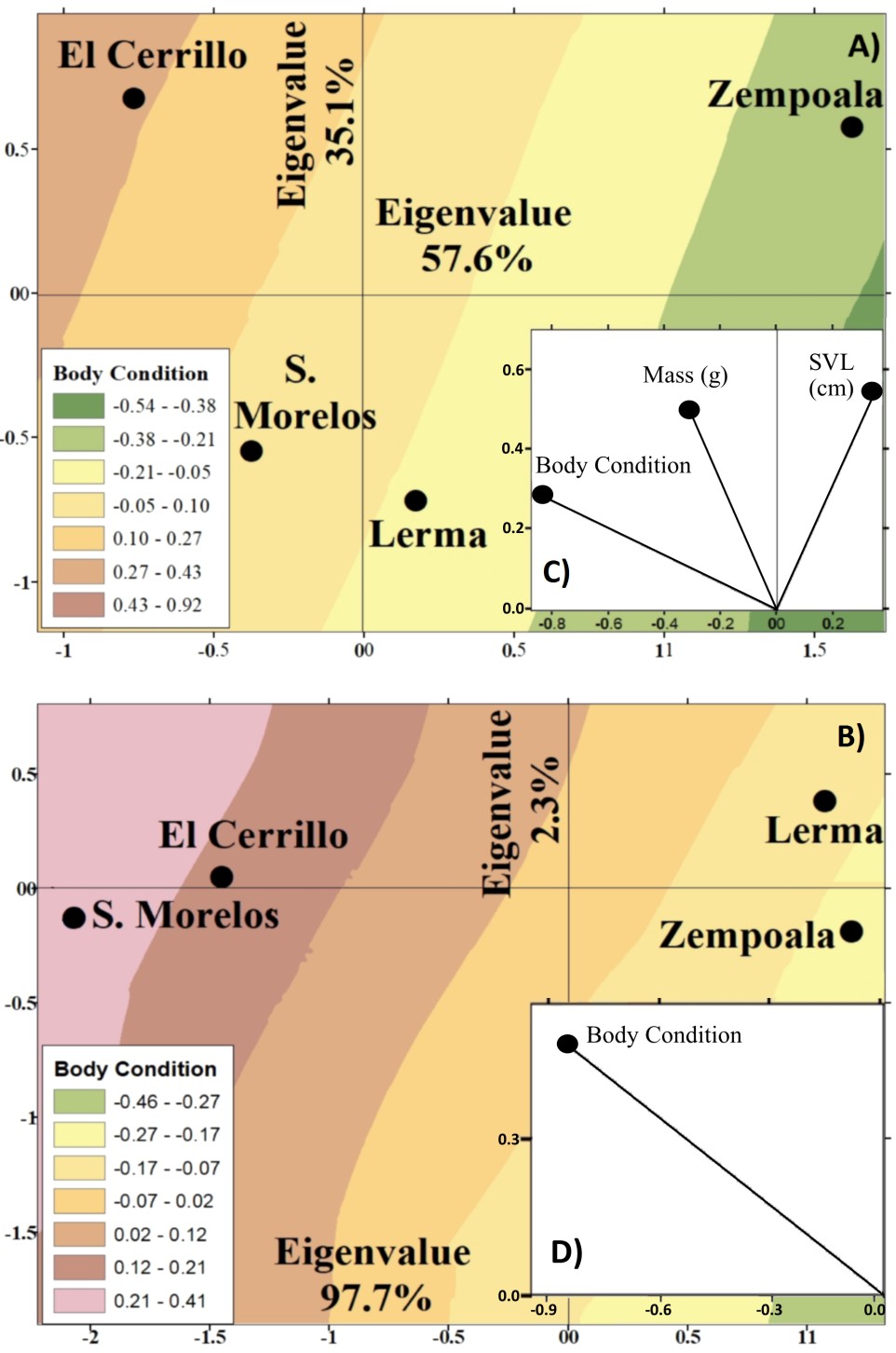

**Figure 4 Canonical position of the centroids of juvenile (A) and adult (B) females of garter snakes *T. scalaris* captured from Lerma, Cerrillo, S. Morelos and Zempoala.** Centroids obtained from a discriminant function analysis and the variables with the greatest discrimination between locations (C and D). Isoclines represent variation of body conditions of snakes in Lerma, Cerrillo, S. Morelos and Zempoala.

**Table 5  Number of stomachs containing each prey taxon ingested by *T. eques* and *T. melanogaster* in the Lerma, Cerrillo and Cuitzeo locations.** Percentages by location in parentheses.

| Prey | Lerma | El Cerrillo | Cuitzeo |
|------|-------|-------------|---------|
| *Thamnophis eques* | | | |
| Fish | 2 (22.2) | 3 (12.5) | 10 (90.9) |
| Leech | 4 (44.4) | 0 | 1 (9.1) |
| Tadpole | 1 (11.1) | 3 (12.5) | 0 |
| Earthworm | 0 | 7 (29.2) | 0 |
| Frog | 2 (22.2) | 5 (20.8) | 0 |
| Mouse | 0 | 6 (25) | 0 |
| *Thamnophis melanogaster* | | | |
| Fish | 30 (88.2) | | 54 (100) |
| Leech | 2 (5.9) | | 0 |
| Tadpole | 0 | | 0 |
| Earthworm | 0 | | 0 |
| Axolotl | 2 (5.9) | | 0 |

The diet of *T. melanogaster* included more prey items at Lerma than in Cuitzeo. At Lerma, the fish were the main prey and leech and axolotl were ingested in similar proportion (Table 5). At Cuitzeo, *T. melanogaster* snakes contained only fishes.

## DISCUSSION

In this study, we provide a heterogeneous pattern of sexual and ontogenic reproductive status variations in body size and condition among populations of three sympatric garter snakes collected in the Central Mexico Highlands over several years. We found: (1) each garter snake species shows good and poor body condition in a variety of locations, (2) juvenile garter snakes show similar body condition between populations, (3) adults show different body conditions between populations, and (4) adults also show sex differences in body condition. Thus, geographical differences in body condition were present in juvenile female *T. eques*, both sexes of adult *T. melanogaster*, and juvenile and adult females of *T. scalaris*.

Addittionally, dietary differences were checked to associated them with body condition differences between the localities. *Thamnophis eques* snakes where fish were the predominant prey (Cuitzeo) had significantly better body condition than snakes that fed on fish and other aquatic, terrestrial and amphibious prey (Lerma and Cerrillo). Overall, although *T. eques* snakes from Lerma and Cerrillo have greater body size than Cuitzeo snakes, adult females were significantly heavier than females from Cuitzeo. On the other hand, *T. melanogaster* snakes where more prey items were ingested (Lerma), had significantly greater body size (SVL and mass) and better body condition (adult males and females) than *T. melanogaster* snakes where fish were the only prey (Cuitzeo).

We hypothesized that body condition of garter snakes would reveal a pattern of geographical variation influenced by ontogenic reproductive status (juvenile, adult), sex, and diet differences among populations. Several problems may confound these

inter- and intraspecific patterns of differences in body condition because each responds to complex interactions between sexual and ontogenic reproductive status with local environmental variables and local resource availability (*Congdon, 1989*; *Shine et al., 2001*). Thus, the differences in body condition between sites may result from differences in local prey availability, dietary quality, or predation efficiency (*Britt, Hicks & Bennett, 2006*), or a complex spatio-temporal interaction that is reflected in micro-geographic diet variation, a pattern common in garter snakes (*Seigel, 1996*). The body condition differences among years and localities within species would be evidence that the patterns found are likely just based on prey availability or climatic constraints on feeding as temporarily fluctuating assimilation rates.

Morphological plasticity induced by diet is extensively documented, especially for natricine snakes (e.g., *Krause, Burghardt & Gillingham, 2003*; *Vincent et al., 2009*; *Hampton, 2013*), and some involve comparisons of snake populations separated by geographic distances. In this study, location and diet were a significant overall factor influencing body size and body condition in garter snakes. The diet has differential effects on *T. eques* and *T. melanogaster*. Both snakes eating fish (Cuitzeo populations) have shorter or lighter relative body sizes, but they respond differentially in their body condition to the piscivorous diet. The generalist *T. eques* have relative best body condition at Cuitzeo, while the specialist *T. melanogaster* apparently does not present significant differences in body condition between the piscivorous population versus other prey. The differences across the two localities may not be strictly due to diet, as is suggested by the fact that juveniles and adult males and females in the two sites show particular differences in any measure of body size (SVL and mass).

Sympatric and closely related species are expected to exhibit a similar body condition due to the ecological similarities that impose common selective pressures, as suggested by the study in closely related and sympatric garter snakes *T. melanogaster* and *T. eques* with similar patterns of neonate body condition (*Manjarrez & San-Roman-Apolonio, 2015*). However, we cannot assume that the garter snakes we studied make similar use of local energy supplies, which may vary according to intra-interspecific competition and available resources (*Congdon, 1989*), especially on prey availability (*Krause, Burghardt & Gillingham, 2003*).

Growth and body condition in snakes may reflect intraspecific competition intensity that would correspond to availability and allocation of energy (*Bronikowski, 2000*; *Bronikowski & Arnold, 1999*; *Blouin-Demers, Prior & Weatherhead, 2002*). This is especially applicable for female garter snakes because they are generally heavier bodied and have greater reproductive energy demands than males (*Naulleau & Bonnet, 1996*; *Shine et al., 2001*; *Blouin-Demers & Weatherhead, 2007*).

For most of the locations in this study, the sex ratio was biased towards females, a common pattern in other species of *Thamnophis* (*Parker & Plummer, 1987*); however, the basic question is whether the variation is true, displaying actual population structure, or is false, reflecting different sexual behavioral traits that can influence catchability of males and females at some locations (*Parker & Plummer, 1987*). The sex ratio of the present locations of the garter snakes studied may not be accurate; in this sense, our conclusions about the

geographical differences of the corporal condition should be considered with caution due to the sexual variability of the body condition between locations.

Ontogenic differences in body condition can result from differential resource use. For example, studies on *T. melanogaster*, *T. eques*, and *T. scalaris* have reported intraspecific differences in the diet of snakes, such as the changing of aquatic invertebrate to terrestrial vertebrate prey between small and large snakes (*Macias-Garcia & Drummond, 1988*; *Manjarrez, Venegas-Barrera & García-Guadarrama, 2007*; *Manjarrez, Macías García & Drummond, 2013*; *Manjarrez, Pacheco-Tinoco & Venegas-Barrera, 2017*). This suggests different trade-off strategies between growth rate and body mass for resource allocation among sites, according to sex (*King, 1989*; *King, 1997*; *Krause, Burghardt & Gillingham, 2003*) and ontogenic reproductive status (*Naulleau & Bonnet, 1996*; *Lind & Beaupre, 2015*). This trade-off has been sparsely studied in neonate snakes (i.e., *Nerodia sipedon* and *Elaphe obsolete*; (*Weatherhead et al., 1999*; *Blouin-Demers & Weatherhead, 2007*).

Another reason for geographic variation in the body condition of juvenile and adult snakes includes geographic variation in the percentage of juveniles and adults in the population. For *T. melanogaster*, 94% of juveniles and 76% of adults were collected from Cuitzeo; while for *T. eques* 70% of juveniles and 77% of adults were collected Lerma. In *T. scalaris* locations, this age bias was less evident, with collection percentages of juveniles ranging from 17% to 31% by location, and 15% to 27% for adults.

According to our results, the models propose different paths for population fitness of each garter snake species assuming the current body condition. In this way, the future scenario responds according to the local geographic variation of each population; however, this prediction is difficult to rely upon because environmental fluctuations can be unpredictable, and changes in the climate, vegetation, topography, and land use variables will reduce the future potential distribution of these three garter snakes, as has been predicted in *González-Fernández et al. (2018)*.

Another important pattern in this study is the interspecific difference of body condition within the same location. For example, in Cuitzeo, the body condition of *T. eques* is good, and in Lerma it is poor, while in *T. melanogaster* the body condition is inverse; poor in Cuitzeo and good in Lerma. This difference could be explained by interspecific differences in resource use and its differential microdistribution. In this sense, *T. eques* is a generalist in its diet, ingesting aquatic and terrestrial prey, while *T. melanogaster* is a specialist ingesting only aquatic prey. The majority of specialist-generalist trade-offs are related with wide ecological traits that result in distinct performance between specialists and generalists (*Drummond, 1983*; *Futuyma & Moreno, 1988*). If these species exploit different foraging environments, it is likely that they are exposed to different environmental conditions. For example, Cuitzeo is a permanent lake that offers a constant aquatic foraging environment for the aquatic specialist *T. melanogaster*, while Lerma is a wetland environment, more suitable for the aquatic-terrestrial *T. eques*, a differential pattern that is reflected in the interspecific differential body condition within both locations. In this sense, the interspecific differences of the body condition can be a reflection of the phenotypic plasticity of both garter snakes, because the geographical difference in the diet is reflected in the local differences of body condition.

The morphological differences found in these studies reflected phenotypic plasticity, rather than genotypic differences, although the relative function of genotype, ontogeny, and sex in the presence of this plasticity could only be inquired through future studies. Also, further exploration, including a larger sample size by local diet, is required.

## CONCLUSIONS

In conclusion, our analyses suggest that traits associated with body condition of sympatric Mexican garter snakes *T. eques*, *T. melanogaster*, and *T. scalaris* in the Central Mexico Highlands reveal a pattern of microgeographical variation among local populations that differ little by ontogenic reproductive status, and therefore sex has little or no influence on body condition in these garter snakes. The diet has differential effects on *T. eques* and *T. melanogaster* in traits associated with body condition.

## ACKNOWLEDGEMENTS

We thank all of the students of the Evolutionary Biology Laboratory for their assistance in the field and laboratory work. Ruthe J. Smith provided comments and corrections regarding the manuscript. EVF is grateful to the graduate program "Maestria en Ciencias Agropecuarias y Recursos Naturales" of "Universidad Autonoma del Estado de Mexico" and to the "Consejo Nacional de Ciencia y Tecnología". Moral support to JM was provides by Carmen Zepeda, Javier and Mariana Manjarrez-Zepeda.

### Funding

This study received funding resources of the Universidad Autónoma del Estado de México (Number 4047/2016SF). The funders had no role in study design, data collection and analysis, decision to publish, or preparation of the manuscript.

### Grant Disclosures

The following grant information was disclosed by the authors:
Universidad Autónoma del Estado de México: 4047/2016SF.

### Competing Interests

The authors declare there are no competing interests.

### Author Contributions

- Erika Valencia-Flores performed the experiments, analyzed the data.
- Crystian S. Venegas-Barrera analyzed the data, prepared figures and/or tables, authored or reviewed drafts of the paper, approved the final draft.
- Victor Fajardo contributed reagents/materials/analysis tools, authored or reviewed drafts of the paper.
- Javier Manjarrez conceived and designed the experiments, analyzed the data, contributed reagents/materials/analysis tools, prepared figures and/or tables, authored or reviewed drafts of the paper, approved the final draft.

## Animal Ethics

The following information was supplied relating to ethical approvals (i.e., approving body and any reference numbers):

This study received the approval of the ethics committee of the Universidad Autónoma del Estado de México (Number 4047/2016SF).

## Field Study Permissions

The following information was supplied relating to field study approvals (i.e., approving body and any reference numbers):

This study received the approval of field permit #07164 from the Secretaria del Medio Ambiente y Recursos Naturales.

## Data Availability

The raw data are available in a Supplemental File.

## Supplemental Information

Supplemental information for this article can be found online at http://dx.doi.org/10.7717/peerj.6601#supplemental-information.

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
