# Peer review of "Microgeographic variation in body condition of three Mexican garter snakes in central Mexico"

_PeerJ, doi:10.7717/peerj.6601_

## Round 0.1 · original submission · Major Revisions

Two of three reviewers have major reservations about your manuscript, and the third is also asking for a number of revisions. Given this, I would like to see a major revision dealing with all of the comments. The revised version will be returned to reviewers and there is no guarantee that your paper will be accepted for publication.

·

Basic reporting

There is some literature that I believe should be strongly considered for citation. Otherwise some key papers are left out that are either conceptually relevant (though with a different species) or directly so because garter snakes are used. For example:

The authors should consult and cite the edited volume Geographic variation in behavior. Perspectives on evolutionary mechanisms (Foster and Endler, eds). Specifically, the chapter by Burghardt and Schwartz, Geographic variations on methodological themes in comparative ethology: A natricine snake perspective. The authors report important results on geographic variation in body size in different populations of garter snakes.


King, R.B. (1989). Body size variation among island and mainland snake populations. Herpetologica, 45, 84-88.

King R.B. (1997). Variation in brown snakes morphology….microgeographic differences. Journal of Herpetology, 31, 335-346.

Krause, M. A., & Burghardt, G. M., & Gillingham, J. C. (2003) Body size plasticity and local variation of relative head and body size sexual dimorphism of garter snakes (Thamnophis sirtalis). Journal of Zoology, 261, 399-407.

NOTE: I can provide pdfs of the Burghardt and Schwartz chapter and Krause et al if the authors would like to contact me directly at [email protected])

Lines 78-79. “Garter Snakes” doesn’t need to be capitalized

Lines 84-87. Confusing as worded. Are the authors stating that previous publications have only focused on males or females, and rarely have both been combined in a single study? Might help to reword this. Also, change “sexual” to “sex” (and on line 251 and wherever else in this context).
Line 237. Is there a missing word on this sentence?
Line 257 typo: reflect result

Experimental design

The study is not experimental in nature. In terms of research design in general, the authors are dealing with natural populations of animals so there is nothing that is controlled. This limits the manuscript in terms of hypothesis testing. However, the field sites provide opportunity for “naturalistic experiments” because the variables of geographic location, sex, and ontogenetic status can be compared and related to existing literature.

One item: Is it statistically appropriate to use the MA linear regression to compare body condition to snakes that differ in overall body size/age (e.g., juveniles and adults)? I do not know, but this might be something to clarify. Do Faulk et al (2017) address this?

Validity of the findings

The measures used are standard for assessing snake body size and condition, and the authors cite and use good resources for statistical selection. Discriminant function analyses are an interesting and insightful way of relating geographic, sex, and ontogeny on body condition. I do not have reservations concerning the validity of the findings.

Additional comments

Abstract: “sympatric species are predicted to exhibit similar body condition when they experience similar environmental conditions”. Is this true? This would seem to assume that these sympatric species are independent of each other, but if there are differential effects of resource use/competition or disease, phylogenetic effects, etc then this would not seem to be a valid prediction. The second sentence of the introduction addresses this, and seems to contradict this statement in the abstract. This comes up again in the discussion lines 261-265. This is confusing because on the one hand the authors are stating that closely related sympatric species should exhibit similar body condition. They then state that this can’t be assumed and that competition over resources, etc might create differences among sympatric species. Which is it? I think this needs to be more fully explored, and with relevant literature cited in more detail. This would help strengthen the discussion section too, which I find is overly focused on the garter snakes studied, and less focused on broader ecological and evolutionary questions.

Lines 304-305. The authors operationally define good or poor condition based on the position of residuals above or below the regression line. Here the terms good and poor are used in reference to populations of animals, not individuals (e.g., “the body condition of T. eques is good”). It might help to remind readers how the statistical operational definition of good/poor translates into the real world (e.g., make it biologically meaningful). Is it this based on mean residuals of condition index was below the regression line? Something else? I think it would help to clarify this.

Lines 303-315 of the discussion are very nice. In fact I think this could be expanded and broadened to include parallel findings with other species (snakes or other animals). The discussion as written is highly focused on rehashing the results, and I would appreciate seeing it broadened.

Reviewer 2 ·

Basic reporting

Writing is clear and professional English.

Literature is well cited and reviewed.

Figures are generally good, but some headings lack enough information.

Experimental design

The hypothesis that 'body condition would vary' is rather simplistic.

Methods are well-described.

Validity of the findings

See below

Additional comments

The authors document spatial and ontogenetic differences in body size and condition, sex ratio among three species of sympatric garter snake.

This paper presented several years of body size data collected from three species of garter snakes. It documents differences in body condition among sites, species etc. This result is not surprising given interspecific differences in diet, ontogenetic changes in prey size etc. It would have been remarkable if no differences were found.

However, no data is presented to offer any explanation of the observed differences. Data on prey abundance, feeding rate, temperature, cloud cover etc. would typically be presented to correlate with differences in condition.

The results of this study don't document evidence of either effect- it merely documents the existence of variation. The second paragraph of the discussion indicates this. The remaining discussion largely recounts previous studies that indicate condition is affected by a combination of environmental factors and local adaptations. But the present study does not add to this body of knowledge.


Some specific comments-
L31. It would be worth clarifying here that the number of years actually sampled ranged from 3-11, over a span of 20 years.

L47-8. Identifying local adaptation also requires concurrent information on prey availability, climate etc, to extricate genetic from environmental plasticity. Body condition of snakes is highly plastic in response to prey quality, availability, individual feeding rates etc. Detecting adaptive divergences in body condition needs to control for these environmental determinants. For example, many of the papers cited throughout the introduction, that document adaptive patterns in life history traits also monitored prey availability, climate etc.

I don't understand the justification for using body condition in DFA along with body mass and SVL. Condition is already derived from mass and SVL. How is body condition providing additional information in a multivariate analysis that includes the variables that were used to calculate it?

L148. If SVL and mass varied dramatically among years, body condition must have too. It would be useful to run this 3-way analysis with body condition as the independent variable, instead of SVL and mass. If condition varies among years within species, sexes, and sites, then it must be temporally labile. This would be evidence that the patterns found are likely just based on prey availability or climatic constraints on feeding / assimilation rates.

L274. Several sites showed significantly skewed sex rations. Could these reflect differences in catchability of males and females at some sites? If so, conclusions regarding sex may be spurious.

Fig 4 Heading appears to be cut off in mid-word 'greates'.

Headings for Fig 4 needs more information. The insets need to be described, for instance. And for Fig 4, explain why there is a ratio of SVL/g in the inset. Why are some sites circled in 4C? Why is there no Figure 4B?

Reviewer 3 ·

Basic reporting

The authors have collected an impressive dataset on morphological attributes of three species of garter snakes from the Mexican highlands. These are important data and help fill a natural history shortfall for these snakes. Unfortunately, the central aim of the paper is poorly defined, and this makes it difficult to follow the flow of logic linking the different sections together. The authors do not use figures to their advantage. There is extensive in-text reporting of results but it takes the form of parenthetical F-statistics and P-values contrasting a difference in some particular morphological trait of interest. There are almost no associated figures for these results, most of which could be concisely presented in one or two multi-paneled plots. The lack of figures makes it quite difficult to discern the patterns the authors are describing. Furthermore, the authors appear to make self-contradictory statements in several places. For example, on line 251 the authors state that “adults also show sexual differences in body condition” but on line 274 the authors state that “we lack evidence to support a difference in body condition based on sex”.

Experimental design

No comment.

Validity of the findings

The discussion is far too speculative given the limits of the data and is only loosely connected to ideas presented in the introductory paragraphs.

---

## Round 0.2 · Minor Revisions

We have now received a second review from one of the original reviewers. This reviewer makes some statistical points that I would like you to consider carefully. If you feel that these points are not valid, please provide me with a well-argued reason to leave your manuscript "as is." In addition, please correct the minor writing errors I have noted in the manuscript. Thanks and I look forward to seeing the next version,

Reviewer 3 ·

Basic reporting

no comment

Experimental design

no comment

Validity of the findings

no comment

Additional comments

Why was individual sex included in the discriminant function analysis (DFA) when the ANOVA revealed that it was not a significant predictor of variation in body condition? Related to this point, why was ontogenetic status included in the DFA when ANOVA was not performed to determine if different ontogenetic stages differed in body condition? Presumably, if you are going to perform and interpret a DFA the groupings that you use should be important predictors of variation in body condition. If sex doesn't explain variation in body condition the differences revealed by DFA are irrelevant, especially because the discriminant functions also include major contributions from SVL and mass. Similarly, without first knowing whether ontogenetic status is an important predictor of variation in body condition how do we know that differences detected by DFA are meaningful?

Related to these points, it is unclear to me why DFA was performed in the first place. The DFA seems to invert the roles of the dependent and independent variables. It uses body condition (as well as SVL and mass) as an independent variable to discriminate among different groupings (sex, location, ontogenetic status). How is this relevant to the central aim of the manuscript laid out in the introduction (and title)?, which in my reading was to determine if the independent variables sex, location, and ontogenetic status are important predictors of variation in individual body condition. Unless there is a compelling reason for keeping it, I suggest dropping the DFA from the manuscript, or at least making it very clear that body condition is being used in two distinct statistical contexts.

---

## Round 0.3 · accepted · Accept

Thanks for making your changes. The article is now Acceptable for publication, but I ask that you one sentence while in production (lines 289-290): "Addittionally, dietary differences were checked to associated they them with body condition differences between the localities." This should read, "Additionally, dietary differences were checked to associate them..." Make sure to note the spelling of "additionally" when you do this.

#